# PCR Detection of *Toxoplasma gondii* in European Wild Rabbit (*Oryctolagus cuniculus*) from Portugal

**DOI:** 10.3390/microorganisms8121926

**Published:** 2020-12-04

**Authors:** Catarina Coelho, Madalena Vieira-Pinto, Anabela Vilares, Maria João Gargaté, Manuela Rodrigues, Luís Cardoso, Ana Patrícia Lopes

**Affiliations:** 1Centro de Ciência Animal e Veterinária (CECAV), Universidade de Trás-os-Montes e Alto Douro (UTAD), 5000-801 Vila Real, Portugal; ccoelho@esav.ipv.pt (C.C.); lcardoso@utad.pt (L.C.); aplopes@utad.pt (A.P.L.); 2Escola Superior Agrária de Viseu, Instituto Politécnico de Viseu, Quinta da Alagoa, 3500-606 Viseu, Portugal; 3Centro de Investigação e de Tecnologias Agroambientais e Biológicas (CITAB), UTAD, 5000-801 Vila Real, Portugal; 4Departamento de Ciências Veterinárias, Escola de Ciências Agrárias e Veterinárias, UTAD, 5000-801 Vila Real, Portugal; mmvcr@utad.pt; 5Departamento de Doenças Infecciosas, Instituto Nacional de Saúde Dr. Ricardo Jorge, 1649-016 Lisboa, Portugal; anabela.vilares@insa.min-saude.pt (A.V.); m.joao.gargate@insa.min-saude.pt (M.J.G.)

**Keywords:** European wild rabbit, *Oryctolagus cuniculus*, polymerase chain reaction, Portugal, surface antigen 2, *Toxoplasma gondii*

## Abstract

Wildlife plays an important role in the epidemiological cycle of *Toxoplasma gondii*. The European wild rabbit (*Oryctolagus cuniculus*) can be a source of infection to wild and domestic hosts, including human beings. Additionally, as an herbivorous animal, the European wild rabbit may also be a sentinel of environmental contamination with *T. gondii* and, consequently, an indicator of the potential transmission of this parasite. The purpose of the present work was to detect *T. gondii* DNA in European wild rabbit from central Portugal, as well as the possible implications for public health. Heart and diaphragm samples were obtained from 28 rabbits hunted in central Portugal. Nested PCR separately amplified the 5′ and 3′ ends of the surface antigen 2 (SAG2) gene. *T. gondii* DNA was detected in 19 out of the 28 sampled animals, resulting in a prevalence of 67.9%. These results show that *T. gondii* infection occurs in European wild rabbit and therefore may pose a potential risk for humans if consumed as raw or undercooked meat. Measures should be taken in order to prevent infection by this zoonotic parasite and for the conservation of wildlife. To the best of our knowledge, this is the first study performed by means of PCR on *T. gondii* in European wild rabbit meat samples.

## 1. Introduction

The European wild rabbit (*Oryctolagus cunniculus*) is native to the Iberian Peninsula [1]. In Iberian ecosystems, it is considered a keystone species, taking into account its importance as a prey of two highly endangered species, the Iberian lynx (*Lynx pardinus*) and the imperial eagle (*Aquila adalbertii*) [2]. In addition to its ecological importance, the wild rabbit is one of the most appreciated small game species [3]. In recent decades, wild rabbit populations have suffered a considerable decrease in the Iberian Peninsula [4]. Several factors have contributed to this decline, including habitat loss, changes in land use, overhunting, natural predation pressure, and two viral diseases, namely, myxomatosis and rabbit hemorrhagic disease [5]. Currently, the species is classified as endangered in its native range by the International Union for Conservation of Nature [6].

The European wild rabbit is an important reservoir of pathogens, including parasites, which can infect domestic animals and even humans [7]. The species is also an excellent choice as a sentinel for evaluating the pathogens circulation in the environment [8].

*Toxoplasma gondii* is an important zoonotic protozoan parasite with worldwide distribution [9]. Cats and other felids are the only known definitive hosts [10,11], while there is a wide range of intermediate hosts, including humans [11,12,13]. Both definitive and intermediate hosts may become infected by one of the following routes [9,13]: (i) Ingestion of food or water contaminated with sporulated oocysts, (ii) ingestion of meat containing infective *T. gondii* cysts, and (iii) vertical transmission.

Foodborne transmission of *T. gondii* is considered the most important route for human infection, and the consumption of raw or undercooked meat or meat products is considered the main risk factor for human infection as well [13,14]. Therefore, wild rabbit meat can be a source for human infections [15].

According to National Hunting Competent Authority (ICNF; personnel communication), 125,117 wild rabbits were hunted during the hunting season of 2018/2019. Although no official data exist about wild rabbit meat consumption in Portugal, it is commonly accepted that all wild rabbit carcasses are used for human consumption, mainly for private use (not placed on the market).

The studies found on *T. gondii* in European wild rabbit have been based only on serological analysis. The presence of antibodies for *T. gondii* has been reported in wild rabbits from Spain (14.2%) [16], Norway and Sweden (21%) [17], the Czech Republic (8%) [18], Scotland (3.3%) [19], and Australia (9.9%) [8]. The current *T. gondii* infection levels in European wild rabbit in Portugal are not really known. To the best of our knowledge, just one recent study in wild rabbits has been reported to date in Portugal, describing a *T. gondii* seroprevalence of 2.8% [20]. The purpose of the present work was to detect *T. gondii* DNA in European wild rabbit from central Portugal, as well as the potential implications for public health. To the best of our knowledge, this is the first study performed on *T. gondii* DNA detection in European wild rabbit meat samples.

## 2. Materials and Methods

### 2.1. Sampling

Twenty-eight wild rabbits were hunted during the hunting season of 2009/2010 (October 2009 to February 2010) in central Portugal. Information about gender, age, and weight was collected. Age was determined by the external assessment of the degree of ossification of the distal ulnar epiphyses and categorized into two age groups: Juvenile (up to 7 months) and adult (over 7 months) [21]. After external examination, the body cavities were opened. The gastrointestinal tract was removed for further parasite evaluation. Tissue samples (heart and diaphragm) were collected and stored frozen at −20 °C until further analysis. Due to the long time period between hunting and sampling, it was not possible to collect blood samples from these animals.

### 2.2. DNA Extraction

DNA extraction was performed using the manual method, with the aid of proteinase K, as described by Miller et al. [22]. To 1 mL of each sample, previously digested by pepsin (DIFCO^®^, Becton, Dickinson and Company, EUA), 4 mL of lysis buffer solution I (in house solution) was added to remove hemoglobin, after which a centrifugation (Heraeus^®^, Multifuge 1, EUA) at 6000 rpm was carried out for 15 min at 4 °C. The resulting pellet was collected, resuspended in 4 mL of lysis buffer I, and again centrifuged under the conditions previously described. To the pellet resulting from the second centrifugation, 1.5 mL of lysis buffer solution II (in house solution), 50 μL of 10% sodium dodecyl sulfate (SDS) (Lauryl, Thermo Fisher, UK), and 26 μL of proteinase K (Life Technologies, Invitrogen, Paisley, U.K.) were added. The samples were then incubated in a water bath at 56 °C overnight. After the incubation period, 400 μL of a 6 M (mol/L) saline solution was added and centrifuged at 5000 rpm for 15 min at 15 °C. The DNA was precipitated by the addition of absolute ethanol (Merck, Germany) at −20 °C and resuspended in 300–500 μL of DNAse/RNAse-free water (GIBCO^®^, Life Technologies, EUA) and frozen at −20 °C until use.

### 2.3. PCR Detection

Considering the low sensitivity of the B1 PCR described in earlier articles concerning animals [23], we opted for performing the Sag2 PCR for all rabbit homogenates. Nested PCR separately amplified the 5′ and 3′ ends of the surface antigen 2 (SAG2) gene [24]. Four PCRs were used to amplify the external and internal sequences of the first and second fragments of the SAG2 gene with external primers F4 GCT ACC TCG AAC AGG AAC AC and R4 GCA TCA ACA GTC TTC GTT GC of the first fragment and F3 TCT GTT CTC CGA AGT GAC TCC and R3 TCA AAG CGT GCA TTA TCG C of the second fragment, as well as the internal primers F GAA ATG TTT CAG GTT GCT GC and R2 GCA AGA GCG AAC TTG AAC AC of the first fragment and F2 ATT CTC ATG CCT CCG CTT C and R AAC GTT TCA CGA AGG CAC AC of the second fragment. The PCR products were visualized in GelRED-stained (Biotarget, Lisbon, Portugal) 2% agarose gel electrophoresis.

Figure 1 presents a flowchart of the methodology used in this study.

### 2.4. Statistical Analysis

The Fisher’s exact test was used to compare proportions of positivity to *T. gondii* between genders (female and male) and ages (juvenile and adult). The Mann-Whitney *U* test compared the weight medians of the positive and negative wild rabbits. A *p*-value ≤ 0.05 was considered statistically significant. Analyses were performed with IBM SPSS Statistics 26^®®^ software (IBM, New York, NY, USA)

## 3. Results

From a total of 28 wild rabbits, *T. gondii* DNA was detected in 19 (67.9%) (Figure 2), suggesting a relevant prevalence in meat samples from the European wild rabbit hunted in the central geographical region of Portugal.

No statistically significant differences were observed between the gender (*p* = 0.371) or age (*p* = 0.097) groups. Nevertheless, a trend toward a statistical association (*p* = 0.097) was observed between the adult age and *T. gondii* positivity (Table 1).

Comparing the median weight of the positive and negative animals, no statistically significant differences were found (*p* = 0.068).

The presence of gastrointestinal helminthes was observed in 26 animals (92.9%) and infection by *Cysticercus pisiformis* in eight rabbits (28.6%). The results from the present study suggest that wild rabbit is an important reservoir of *T. gondii* and other parasites and can be an appropriate choice as a sentinel to evaluate environmental contamination, taking into account that it is an herbivorous animal.

## 4. Discussion

Little is known about *T. gondii* circulation in the wild rabbits of Portugal. The available data are focused only in the southeast of the country, with a *T. gondii* seroprevalence of 2.8% (*n* = 36) [20]. In other countries, most of the studies found on *T. gondii* in European wild rabbit are also focused on serological analysis, with seroprevalence levels varying from 3.3% in Scotland [19] to 21.0% in Norway and Sweden [17]. According to the authors’ knowledge, no studies were found reporting *T. gondii* DNA in muscle samples from wild rabbits. As such, it is not possible to comparatively evaluate the results found in the present study.

Although no differences were observed in *T. gondii* positivity between gender and age groups, a trend was noted toward an association between adult age and *T. gondii* positivity. Serological studies in other wild animals have reported an increase in *T gondii* seropositivity in older age groups [25,26]. However, once again, no studies were found on wild rabbits.

PCR is widely used for the molecular detection of *T. gondii* due its high sensitivity [27]. Different gene loci and markers have been used for the detection of *T. gondii*, and one of the most used for this purpose is the B1 gene [28]. However, to improve the sensitivity of the method, we used the SAG2 gene, since we previously had better results with this gene [23]. The PCR results, when we used the SAG2 gene, showed a sensitivity of 87.5% [29] and a specificity of 100% [30].

In wild rabbits, *T. gondii* infection has been documented as a subclinical disease [8,16,19], in contrast to domestic rabbits, in which fatal cases of toxoplasmosis have been described [9,31]. However, parasitic infection has detrimental effects on host health and fitness, and debilitated animals become easy prey for their predators [32]. Wild rabbits are a common prey of many wild carnivores—some of them being endangered species, such as the Iberian lynx and the imperial eagle. In Portugal, the European wild rabbit may be an important source of *T. gondii* infection for the Iberian lynx and other wild carnivores such as foxes, and therefore can compromise wildlife conservation. As an herbivorous animal, infection with *T. gondii* in wild rabbits may also be a marker of environmental contamination by sporulated oocysts [8,9,13].

The presented results reveal that *T. gondii* infection occurs in European wild rabbit from Portugal and may, therefore, pose a potential risk, not only to the hunters and their families, but also to other people who consume this meat raw or undercooked. In Portugal, wild rabbit meat is widely consumed mainly by hunters and their families. Moreover, hunters have an important role in the life cycle of *T. gondii* when they are handling or eviscerating carcasses. The potentially infected viscera and carcasses not of interest are left behind, posing a threat by means of *T. gondii* dissemination due to scavenging by carnivorous or omnivorous susceptible animals [33]. To prevent further infection, the correct disposal of viscera and carcasses should be promoted as Veterinary Public Health and Wildlife Conservation measures, in order to interrupt the life cycle of *T. gondii*. Likewise, mitigation of this and other zoonoses could be optimized if hunters were properly trained and informed as is recommended by European Regulation (EC) No. 853/2004.

In this work, we presented the first data concerning *T. gondii* DNA detection in European wild rabbit meat samples. Since it is a zoonotic parasite, we would like to emphasize the importance of implementing preventive measures in order to mitigate human infection. Given that no genotyping analysis was carried out in this study, additional studies should be developed, with a larger number of samples, in order to identify the circulating *T. gondii* strains to better understand the epidemiological importance of this zoonosis in wild rabbits from Portugal. Nevertheless, the importance of the present study must be stressed due to the zoonotic character of the parasite, as well as the originality of the parasite detection in wild rabbits in Portugal and even internationally. This is a preliminary work, but it opens the door to future research in this area, providing additional and updated information on this subject.

## Figures and Tables

**Figure 1 microorganisms-08-01926-f001:**
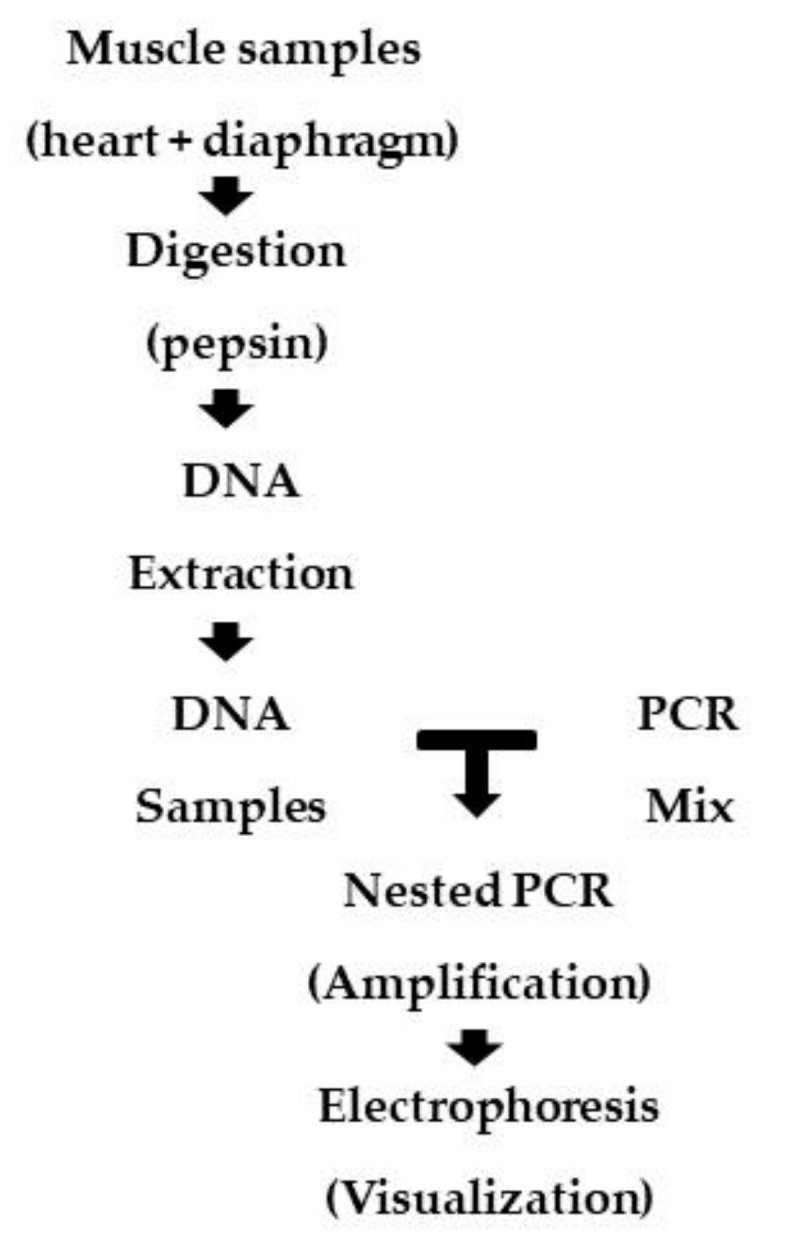
Flowchart of the methodology steps.

**Figure 2 microorganisms-08-01926-f002:**
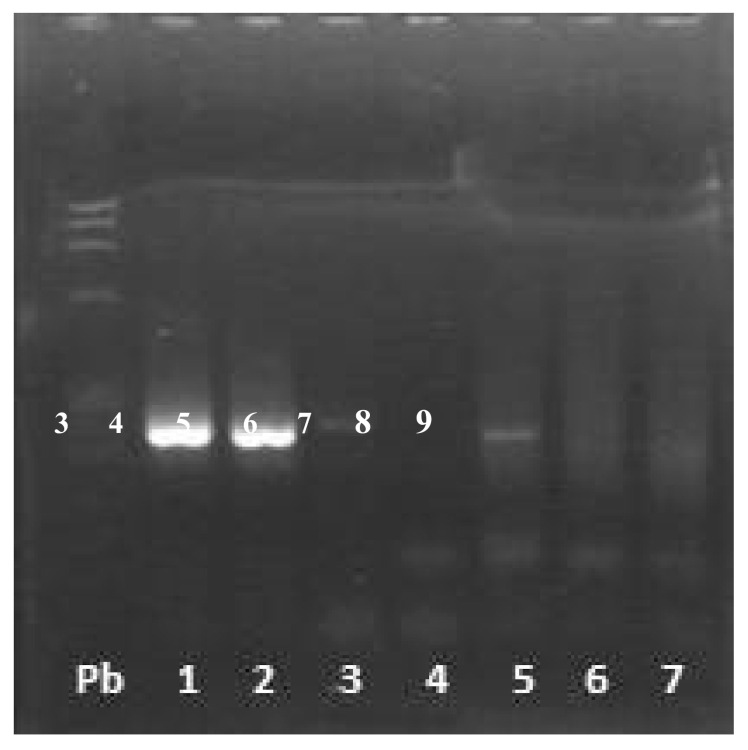
Nested PCR for the first fragment of the surface antigen 2 (SAG2) gene.

**Table 1 microorganisms-08-01926-t001:** Wild rabbit *T. gondii* positivity by gender and age groups.

	Wild Rabbit (*n*)	PCR Positive (*n*)	Prevalence (%)	*p*-Value
**Gender**				0.371
Male	8	4	50.0	
Female	20	15	75.0	
**Age groups**				0.097
Juvenile (≤7 months)	9	4	44.4	
Adult (>7 months)	19	15	78.9	
**Total**	**28**	**19**	**67.9**

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
