# Peer review of "PCR Detection of Toxoplasma gondii in European Wild Rabbit (Oryctolagus cuniculus) from Portugal"

_microorganisms, 2020, doi:10.3390/microorganisms8121926_

Round 1
Reviewer 1 Report
To ,,Introduction" - please add some information about of human population habits of eating rabbit meat, in your region.
To ,,Introduction" - please add some seroprevalence data of T. gondii in wild rabbit, from other studies in the world.
Have you collected also blood samples from the wild rabbits? Have you done some serological screening tests for T. gondii? Can you compare the blood screening of the wild rabbits with the PCR results?
Add some pictures of the PCR results, from electrophoresis.
Details about genotyping....
Comparisons with the references from GenBank....
Present on the ,,Discussions" you we used the PCR method, the sensibility and sensitivity of the method for this pathogen.
The conclusions have to be highlighted, presented more clear...
The results would be valuable if the work were be improved.
Author Response
Dear Reviewer
We would like to thank you for making excellent suggestions to improve the quality of our manuscript. We have taken into consideration all of your comments. Below we have answered (A) the questions (Q) that you raised.
Q1. To ,,Introduction" - please add some information about of human population habits of eating rabbit meat, in your region.
A1. Revised as requested. The following paragraph has been included (line 63): “According to National Hunting Competent Authority (ICNF, personnel communication), 125117 wild rabbits were hunted during the hunting season of 2018/2019. Although no official data exist about wild rabbit meat consumption in Portugal, it is commonly accepted that all wild rabbit’s carcasses are used for human consumption, mainly for private use (not placed on the market).”
Q2. To ,,Introduction" - please add some seroprevalence data of T. gondii in wild rabbit, from other studies in the world.
A2. Revised as requested. We have inserted the paragraph (line 67): "The studies found on T. gondii in European wild rabbits have been based only on serological analysis. The presence of antibodies to T. gondii has been reported in wild rabbits from Spain (14.2%) (Almeria et al., 2004), Norway and Sweden (21%) (Kapperud, 1978), Czech Republic (8%) (Hejlíček et al., 1997), Scotland (3.3%) (Mason et al., 2015) and Australia (9.9%) (McKenny et al., 2020)."
Q3. Have you collected also blood samples from the wild rabbits? Have you done some serological screening tests for T. gondii? Can you compare the blood screening of the wild rabbits with the PCR results?
A3. Unfortunately we couldn't collect blood samples and consequently we don't have serological results. The animals were killed by hunters and the time laps between killing and sampling was too long. Because of that it's hard (almost impossible) to collect blood from these animals. Based on this, the following sentence has been included on material and methods (line 86): “Due to the long time period between hunting and sampling, it was not possible to collect blood samples from these animals.”
Q4. Add some pictures of the PCR results, from electrophoresis.
A4. Revised as requested. We have inserted one figure about Nested PCR for the first segment of the gene SAG2 (Figure 1)
Q5. Details about genotyping....
A5. This study was performed in 2010 and due to the small amount of DNA, the genotyping wasn't done at that time. The samples are no longer available, so we aren't able to include details about genotyping as you asked, although we understand their importance. This limitation was underlined in the final part of the manuscript (line 182): “Given that no genotyping analysis was carried out in this study, additional studies should be developed, with a larger number of samples, in order to identify the circulating T. gondii strains to better understand the epidemiological importance of this zoonosis in wild rabbits from Portugal.”
Q6. Comparisons with the references from GenBank....
A6. As previously explained we don't have any information about the sequences and therefore we can't compare with the references from GenBank. The same as the answer above.
Q7. Present on the ,,Discussions" you we used the PCR method, the sensibility and sensitivity of the method for this pathogen.
A7. Revised as requested. We have inserted the paragraph (line 155): “PCR is widely used for molecular detection of T. gondii due is high sensitivity (Su et al., 2009). Different gene loci and markers have been used for detection of T. gondii and one of the most used for this purpose is B1 gene (Jones et al., 2000). However, to improve the sensitivity of the method, we used the SAG2, since we previously had better results to this gene (Vilares et al., 2014). The PCR, when we used SAG2 gene, showed a sensitivity of 87.5% (Ramírez et al., 2017) and specificity of 100% (Vethencourt et al., 2019).”
Q8. The conclusions have to be highlighted, presented more clear...
A8. We apologize for not having clarified the conclusions. We have now rephrased to read as (line 180): “In this work we present the first data concerning T. gondii DNA detection in European wild rabbit meat samples. Since it is a zoonotic parasite, we would like to emphasize the importance of implementing preventive measures in order to mitigate human infection. Given that no genotyping analysis was carried out in this study, additional studies should be developed, with a larger number of samples, in order to identify the circulating T. gondii strains to better understand the epidemiological importance of this zoonosis in wild rabbits from Portugal. Nevertheless, the importance of the present study must be stressed due to the zoonotic character of the parasite, the originality of parasite detection in wild rabbits, in Portugal and even internationally. It is a preliminary work, but it opens the door to future research in the area, providing additional and updated information on this subject.”

Reviewer 2 Report
The purpose of the present work was to 59 detect T. gondii DNA in European wild rabbit from Central Portugal, and potential implications for public health.
Comments:
- A flowchart of the paper will be welcomed and very useful for readers.
- Keywords: no acronyms in keywords; explain these or replace these
- Introduction: more data regarding Toxoplasmosis in rabbits will also be welcomed.
- Discussion section has to be expanded and including more similar research for comparison. The authors also can present the potential perspective of their experimental study.
- I suggest mentioning the limitations and pitfalls of this paper.
Author Response
Dear Reviewer
We would like to thank you for making excellent suggestions to improve the quality of our manuscript. We have taken into consideration all of your comments. Below we have answered (A) the questions (Q) that you raised.
Q1. A flowchart of the paper will be welcomed and very useful for readers.
A1. In order to meet the reviewers' expectations regarding the inclusion of a flowchart, the authors cordially request clarification regarding the flowchart theme and in which chapter of the article it should be included.
Q2. Keywords: no acronyms in keywords; explain these or replace these.
A2. Revised as requested. We replace the acronym PCR in keywords (line 37) by "Polymerase Chain Reaction" and SAG2 by "Surface Antigen 2".
Q3. Introduction: more data regarding Toxoplasmosis in rabbits will also be welcomed.
A3. Revised as requested. We have inserted the paragraph (line 67): We have inserted the paragraph (line 67): "The studies found on T. gondii in European wild rabbits have been based only on serological analysis. The presence of antibodies to T. gondii has been reported in wild rabbits from Spain (14.2%) (Almeria et al., 2004), Norway and Sweden (21%) (Kapperud, 1978), Czech Republic (8%) (Hejlíček et al., 1997), Scotland (3.3%) (Mason et al., 2015) and Australia (9.9%) (McKenny et al., 2020)."
Q4. Discussion section has to be expanded and including more similar research for comparison. The authors also can present the potential perspective of their experimental study.
A4. According to author´s knowledge, no studies were found reporting T. gondii DNA in muscle samples from wild rabbit. In this way, it is not possible to comparatively evaluate the results found in the present study. We have inserted the following paragraph (line 180) “In this work we present the first data concerning T. gondii DNA detection in European wild rabbit meat samples.”
Q5. I suggest mentioning the limitations and pitfalls of this paper.
A5. Revised as requested. We have inserted the following sentences (line 86): “Due to the long time period between hunting and sampling, it was not possible to collect blood samples from this animals.” and (line 182): “Given that no genotyping analysis was carried out in this study, additional studies should be developed, with a larger number of samples, in order to identify the circulating T. gondii strains to better understand the epidemiological importance of this zoonosis in wild rabbits from Portugal. Nevertheless, the importance of the present study must be stressed due to the zoonotic character of the parasite, the originality of parasite detection in wild rabbits, in Portugal and even internationally. It is a preliminary work, but it opens the door to future research in the area, providing additional and updated information on this subject.”

Round 2
Reviewer 1 Report
The author made the correction recommended
Author Response
Dear Reviewer
We would like to thank for your suggestions and comments that have improve the quality of our manuscript.
Reviewer 2 Report
The authors addressed other comments, only one comment need to be address.
Regarding the author's question: the authors cordially request clarification regarding the flowchart theme and in which chapter of the article it should be included.
A flowchart can be added in the material and methods part showing the steps of the study.
Author Response
Dear Reviewer
We would like to thank you the clarification regarding the flowchart. We have taken into consideration your comment. Below we have answered (A) the question (Q) that you raised.
Q1. A flowchart can be added in the material and methods part showing the steps of the study.
A1. Revised has requested. We have inserted a flowchart resuming the methodology steps (Figure 1), hoping it will meet your expactations.

This manuscript is a resubmission of an earlier submission. The following is a list of the peer review reports and author responses from that submission.